# Parafibromin Is Highly Expressed in Hepatocellular Carcinoma and Its Expression Correlates with Poor Prognosis

**DOI:** 10.3390/jcm11071773

**Published:** 2022-03-23

**Authors:** Min-Kyung Kim

**Affiliations:** Department of Pathology, College of Medicine, Dongguk University, Gyeongju 38066, Korea; minkyungk76@naver.com; Tel.: +82-54-703-7810

**Keywords:** hepatocellular carcinoma, parafibromin protein, hepatitis virus, prognosis, immunohistochemistry

## Abstract

Hepatocellular carcinoma (HCC) is the second leading cause of cancer-related death worldwide. Much progress has been made regarding the understanding of hepatocarcinogenesis, yet the long-term survival rate of HCC patients remains poor. Recent efforts have shown parafibromin has a pathologic role in many human cancers, but little is known about the effects of parafibromin in HCC. This study aimed to investigate the pattern of parafibromin expression and its clinicopathologic significance in human HCC. Immunohistochemical analysis of HCC and matched non-tumor liver tissues from 50 HCC patients showed that the nuclear expression of parafibromin was higher in HCC tissues (50/50 cases) than in non-tumor liver tissues (17/50 cases). Moreover, elevated parafibromin expression was found to be significantly correlated with the presence of microvascular invasion (*p* = 0.017), hepatitis virus infection-induced occurrence (*p* = 0.005), and poorer tumor differentiation (Edmondson–Steiner grade; *p* = 0.000). Kaplan–Meier analysis showed that HCC patients with elevated parafibromin expression had poorer recurrence-free (*p* = 0.014, log-rank test = 6.079) and overall survival (*p* = 0.036, log-rank test = 4.414). These findings indicate parafibromin may be related to the pathogenesis of HCC and a potential prognostic marker for HCC patients after hepatectomy.

## 1. Introduction

Hepatocellular carcinoma (HCC) is the most common primary liver cancer and the second leading cause of cancer-related death worldwide [1]. Hepatitis B virus (HBV), hepatitis C virus (HCV), alcoholic steatohepatitis, and non-alcoholic steatohepatitis (NASH) are the major causative factors of HCC development [2,3]. In most cases, HCC develops in chronic liver disease that provides the basis for premalignant lesions such as dysplastic foci and dysplastic nodules through multistep hepatocarcinogenesis [4]. During the carcinogenetic process, the vascular remodeling, such as neoangiogenesis and sinusoidal capillarization, and the gradual loss of the proteins responsible for the uptake and excretion of gadolinium-chelates are fundamental to the diagnosis of HCC with dynamic computed tomography (CT) or magnetic resonance imaging (MRI) and gadoexetic acid-enhanced MRI [4,5,6]. Much progress has been made with respect to the understanding of hepatocarcinogenesis, yet the long-term survival rate of HCC patients remains poor. Thus, it is of considerable importance that research aimed at identifying molecular mechanisms, therapeutic targets, and prognostic biomarkers of HCC be energetically pursued.

Chronic inflammation caused by chronic viral hepatitis, alcohol abuse, and metabolic liver disease underlies the pathogenesis of HCC and leads to a continuous process of immune-mediated destruction and compensatory regeneration [3,7,8]. During this process, genetic abnormalities accumulate, and these contribute to cancer initiation and progression [9,10]. The Wnt/β-catenin signaling pathway is an attractive candidate mediator of chronic hepatic inflammation, and parafibromin (cell division cycle 73, CDC73) binds stably to β-catenin in the nuclei of hepatocytes to form parafibromin/β-catenin complex, which induces the expressions of *Wnt* target genes such as *cyclnD1* and *c-myc* [11,12].

Parafibromin is a protein encoded by the *HRPT2/CDC73* gene located on chromosome 1q31 and a component of RNA polymerase ΙΙ-associated factor (PAF) complex, which is involved in the regulation of cell proliferation, transcription, and histone modification [13,14,15,16,17]. Recent studies have reported associations between parafibromin expression and prognosis for several tumor types, including hyperparathyroidism-jaw tumor (HPT-JT), and parathyroid, breast, lung, gastric, and colorectal cancer [13,14,17,18,19,20,21,22,23].

However, little is known about the activities of parafibromin in hepatocarcinogenesis. More specifically, patterns of parafibromin expression and its prognostic value have not been investigated in human HCC tissues. Therefore, this retrospective study was undertaken to document parafibromin expression patterns and their clinicopathologic significance in patients with HCC.

## 2. Materials and Methods

### 2.1. Patient Clinicopathologic Features and Tissue Microarray Construction

The medical records and pathologic reports of 50 HCC patients (42 males and 8 females, mean overall age 55.0 years, range: 31 to 73 years) who underwent hepatectomy as an initial treatment at Keimyung University, Dongsan Medical Center, from January 2003 to December 2004 were reviewed. Mean follow-up was 56.7 months (range: 1.1 to 137.7 months). Twenty-two patients (44.0%) expired during follow-up and seven patients (14.0%) were lost to follow-up. Tumor size was defined as maximum surgical tumor specimen diameter. Numbers of tumor nodules and vascular invasions were determined by reviewing abdomen and pelvic dynamic CT findings, operation records, and surgical pathology reports. Tumor differentiation was classified using the Edmonson–Steiner grading system (grade 1–4), and overall survival time was defined as time between hepatectomy and disease-related death or last follow-up.

Tissue microarrays (TMAs) were constructed using a manual tissue microarray unit (Quick Ray, Unitma, Seoul, Korea). In brief, formalin-fixed, paraffin-embedded tissue blocks and corresponding hematoxylin and eosin (H&E)-stained slides were reviewed and selected for TMA sampling by two pathologists. A punch (3.0 mm tip diameter) was used to obtain representative tumor and matched non-tumorous liver tissues from individual donor tissue blocks, and these were used to construct paraffin blocks using an arraying device (Unitma, Seoul, Korea). Two tumor cores and two matched non-tumor cores per case were used to construct TMAs.

### 2.2. Immunohistochemical Staining

TMA sections (4 μm) were H&E stained for pathological reassessment and subjected to immunohistochemical (IHC) staining using an autostainer (LV360-2D) and an UltraVision LP Kit (TL-060-HD; Lab Vision Corporation, Fremont, CA, USA). The primary antibody used was mouse anti-parafibromin monoclonal antibody (SC-33638, diluted 1:50, Santa Cruz Biotechnology, Dallas, TX, USA).

The whole slides were scanned using a Panoramic MIDI slide scanner equipped with iSolution DT software. Parafibromin positivity was scored based on nuclear staining intensities and percentages of positive hepatocytes as follows: −, absent; +, weak; ++, focally strong in <50% of cells; +++, diffusely strong expression in ≥50% of cells; and ++++, diffusely strong nuclear and cytoplasmic expression with accompanying strong cytoplasmic positivity. This definition of ‘weak’ corresponded with parafibromin being mainly expressed in nucleoli, and ‘strong’ with it being expressed throughout nuclei. Images were captured using a DP 70 optical microscope equipped with a digital camera (Olympus, Tokyo, Japan).

### 2.3. Statistical Analysis

Data are expressed as means ± standard deviations (SDs) for continuous variables and as frequencies and percentages for categorical variables. Correlations between patient clinicopathologic variables and parafibromin expressions in individual tissue sections were analyzed using Pearson’s chi-square test for categorical variables. The Kruskal–Wallis test was used to determine the nature of the correlation between parafibromin expressions and patient age. Survival analysis was performed using a Kaplan–Meier survival curve and the log-rank test. HCC-specific recurrence-free survival (RFS) was calculated from date of hepatectomy to date of HCC recurrence, death, or last follow-up. Overall survival (OS) was calculated from date of hepatectomy to date of death or last follow-up. The analysis was performed using IBM SPSS Statistics Ver. 25.0 (IBM Corp., Armonk, NY, USA), and statistical significance was accepted for two-sided *p* values < 0.05.

## 3. Results

### 3.1. Parafibromin Expression Patterns

The nuclear expression of parafibromin was significantly different in HCC tissues and matched non-tumor liver tissues. In non-tumor tissues, parafibromin expression was absent (33/50 cases) or weak (15/50 cases) in the nuclei of hepatocytes. However, in two cases, focally strong nuclear staining was observed in hepatocytes around heavy inflammatory infiltrates (2/50 cases).

In HCC liver tissues, all hepatocytes showed various degrees of parafibromin expression throughout whole nuclei. Weak nuclear staining in tumor tissues was observed in 8 of the 50 cases, focally strong staining in 21, and diffusely strong staining in 21. Notably, accompanying strong cytoplasmic immunoreactivity was observed in 7 of the 21 cases with diffusely strong nuclear staining. Intriguingly, high nuclear parafibromin expression corresponded to poorer tumor differentiation (Figure 1).

### 3.2. Clinicopathologic Significance of Parafibromin Expression

Pearson’s correlation coefficients were used to assess associations between clinicopathologic parameters and parafibromin expression. Elevated parafibromin expression was significantly related to the presence of microvascular invasion (*p* = 0.017), hepatitis virus infection-induced occurrence (*p* = 0.005), poorer tumor differentiation (Edmondson–Steiner grade; *p* = 0.000), and 5-year survival (*p* = 0.005). Associations found between parafibromin expression and clinicopathologic parameters are summarized in Table 1.

When HCC patients were divided into low (*n* = 29) and high (*n* = 21) parafibromin expression groups, high parafibromin expression correlated with poorer prognosis. Kaplan–Meier survival curve analysis showed that tumors with high parafibromin expression were associated with poorer recurrence-free survival (RFS) (*p* = 0.014, log-rank test = 6.079) and overall survival (OS) (*p* = 0.036, log-rank test = 4.414) (Figure 2).

## 4. Discussion

HCC is the most common primary liver cancer and the second leading cause of cancer-related mortality worldwide [1]. Human HCC is an inflammation-induced cancer, and its primary causes are HBV, HCV, alcoholic steatohepatitis, and NASH [2,3]. Chronic viral hepatitis, alcohol abuse, and metabolic liver diseases cause chronic inflammation, which leads to continuous cycles of hepatocytes destruction and regeneration, and during compensatory hepatic regeneration, genetic abnormalities accumulate and contribute to the pathogenesis and progression of HCC [3,7,8,9,10]. The Wnt/β-catenin signaling pathway is considered an attractive candidate mediator of chronic inflammation and a critical player in HCC progression [11]. Although the nuclear accumulation of β-catenin is restricted to late-stage HCC, its nuclear accumulation is observed in 40–70% of HCCs [24]. According to a recent review paper on interactions between etiological factors and components of Wnt/β-catenin signaling in HCC, regulatory hepatitis B viral X protein (HBx) plays an important role in Wnt/β-catenin signaling activation in infected hepatocytes and acts as a multifunctional regulator of HBV gene transcription and replication [24]. Furthermore, HCV non-structural 5A (NS5A) protein and structural E2 protein have been reported to activate Wnt/β-catenin signaling via the actions of Src homology region 2 domain-containing phosphatase-2 (SHP-2) and parafibromin, and thereby to induce target gene expression [12,24]. In the nucleus, β-catenin stably binds to parafibromin (cell division cycle, CDC73), and the parafibromin/β-catenin complex formed induces the expression of *Wnt* target genes such as *cyclinD1* and *c-myc*, which finally results in HCC [12,25,26]. Thus, it appears parafibromin might act as an oncoprotein. As was expected, the present study showed the nuclear expression of parafibromin was increased in HCC tissues as compared with matched non-tumor liver tissues (Figure 1) and that this upregulation was significantly related to HBV and HCV infection-induced HCC (*p* = 0.005), which raises questions regarding the mechanics of parafibromin involvement in hepatitis virus-induced HCC.

Parafibromin is encoded by *HRPT2/CDC73* gene and a known tumor suppressor [13,14,18,19,27]. Dysfunction of parafibromin can disrupt cell cycle regulation by promoting cell progression to the S phase, which is associated with tumorigenesis of HCC cells [16,28]. According to A. Porzionato et al., in the only immunohistochemical study performed to date on parafibromin expression in human tissues, parafibromin is normally expressed in parathyroid, adrenal gland, skin, heart, bone marrow, kidney, and liver, and its expression levels and cellular locations are dependent on tissue and cell types. Furthermore, they reported that hepatocytes in normal liver show strong parafibromin immunostaining in nuclei and cytoplasm [29]. Knowledge of parafibromin expression patterns in normal organs and tissues would be expected to provide clues as to whether it acts as a tumor suppressor or oncoprotein during tumorigenesis because its lost or diminished expression in organs and tissues during tumorigenesis would indicate that it functions as a tumor suppressor. However, the present study showed parafibromin expression was much higher in hepatocytes in HCC tissues than in normal hepatocytes and hepatocyte nuclei rather than in cytoplasm (Figure 1). Nonetheless, the upregulation of parafibromin in HCC tissues suggests that it plays a positive role in HCC tumorigenesis.

Over recent years, complete or partial loss of parafibromin expression has been reported in HPT-JT, parathyroid, colorectal, and lung cancer, and its expression has been negatively associated with tumor aggressiveness and poor prognosis [18,19,21,23,30]. However, bioinformatics analysis showed that *CDC73* mRNA expression is higher in gastric, lung, breast, and ovarian cancer than in normal tissues, and positive correlations were reported between *CDC73* mRNA expression and overall and progression-free survival rates in gastric cancer, but that negative correlations were observed in lung, breast, and ovarian cancer [27]. Accordingly, it is not surprising that debate continues as to whether parafibromin acts as a tumor suppressor or an oncoprotein. Although the role of parafibromin has been studied in many tumor types, little is known about its role in HCC. To determine the role of parafibromin in HCC, the present study investigated associations between its expression and clinicopathologic parameters in HCC patients. The nuclear expression of parafibromin was found to be markedly higher in HCC tissues than in matched non-tumor liver tissues. Furthermore, elevated parafibromin expression was found to be closely correlated with the presence of microvascular invasion (*p* = 0.017) and poorer tumor differentiation (Edmondson–Steiner grade; *p* = 0.000) and 5-year survival (*p* = 0.005) (Table 1). In addition, elevated parafibromin expression was associated with poorer RFS (*p* = 0.014, log-rank test = 6.079) and OS (*p* = 0.036, log-rank test = 4.414) (Figure 2). Taken together, it appears parafibromin can act as an oncoprotein in liver and contribute to progression to HCC. This topic requires further investigation.

Various clinical/imaging, morphological, and molecular features are used to predict the prognosis of patients with HCC. Immunohistochemical expression of cytokeratin 19 (CK19) in HCC is related to worse prognosis with more frequent vascular invasion, poor differentiation, higher recurrence rates, and higher rates of lymph node metastasis [31]. In addition, CK19-positive HCCs have higher recurrence of HCC after radiofrequency ablation [32] and higher rates of resistance to transarterial chemoembolization [33]. As CK19 expression is a prognostic factor in HCC, parafibromin expression could be used as a marker of more aggressive behavior and as a prognostic marker after hepatectomy. In clinical practice, the evaluation for parafibromin immunostaining can be applied to tissue obtained by liver biopsy, which is helpful for patient management. In addition, in countries where HBV and HCV infections are major contributors to the prevalence of HCC, parafibromin expression might be a useful predictive marker of HCC development, tumor differentiation, and patient survival.

## 5. Conclusions

The present study is the first to evaluate parafibromin expression in human HCC tissues. Parafibromin was found to be highly expressed in HCC, and its expression was more frequent in HCC tissues with microvascular invasion of poor hepatocyte differentiation and in patients with HBV or HCV infection. Notably, elevated parafibromin expression was negatively correlated with RFS and OS. These findings indicate parafibromin should be considered an oncoprotein and as a potential prognostic marker. Despite the effects of parafibromin overexpression observed in the present study, it is not clear whether parafibromin is regulated by Wnt/β-catenin signaling. Nevertheless, parafibromin presents an attractive developmental target for future HCC treatments.

## Figures and Tables

**Figure 1 jcm-11-01773-f001:**
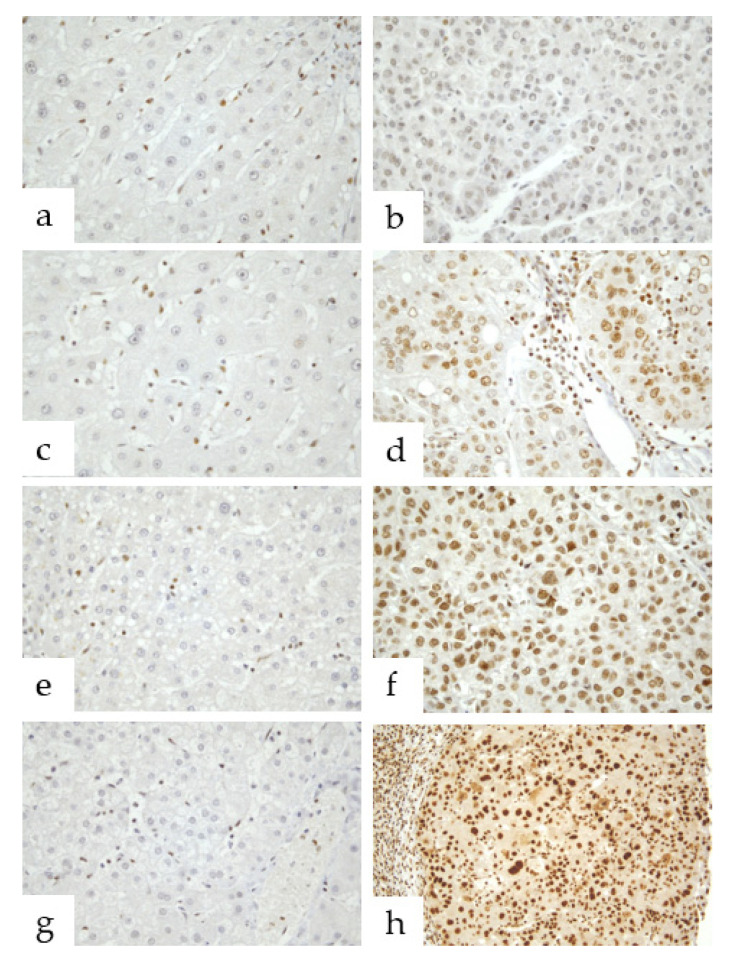
Immunohistochemical patterns of parafibromin expression in human liver tissues. (**a**,**c**,**e**,**g**) In matched non-tumor liver tissues, parafibromin was not expressed in hepatocytes at all (×400). (**b**,**d**,**f**,**h**) In representative images of tumor tissues, parafibromin was expressed weakly in the nuclei of some hepatocytes ((**b**), ×400), strongly in <50% of hepatocyte nuclei ((**d**), ×400), strongly in ≥50% of hepatocyte nuclei ((**f**), ×400), and strongly throughout the nuclei of all hepatocytes in the presence of cytoplasmic expression ((**h**), ×400).

**Figure 2 jcm-11-01773-f002:**
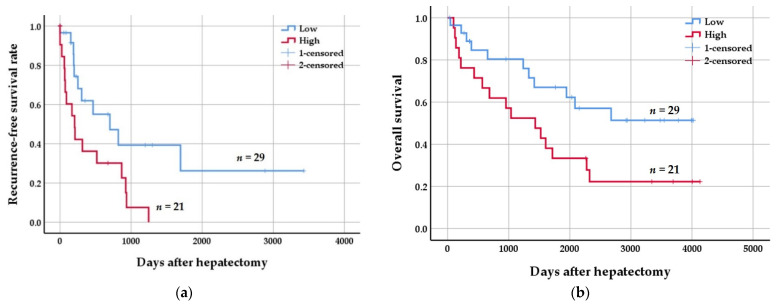
Kaplan–Meier curves of recurrence-free (RFS) and overall (OS) survival rates of hepatocellular carcinoma (HCC) patients according to parafibromin expression status. (**a**) The low parafibromin expression group had better RFS than the high expression group (*p* = 0.014, log rank = 6.079). (**b**) The low parafibromin expression group had better OS than the high expression group (*p* = 0.036, log rank = 4.414).

**Table 1 jcm-11-01773-t001:** Associations between parafibromin expression and clinicopathologic parameters in patients with HCC ^1^.

Clinicopathologic Parameters	Parafibromin Expression				*p* Value
	+	++	+++	++++	
	(*n* = 8)	(*n* = 21)	(*n* = 14)	(*n* = 7)	
Age (mean ± SD ^2^), years	53.8 ± 10.0	57.9 ± 6.9	49.6 ± 12.5	58.0 ± 10.0	0.152
Gender					0.467
Male	8	17	12	5	
Female	0	4	2	2	
Tumor size					0.978
≤3 cm	2	5	5	1	
>3 cm	6	16	9	6	
Tumor multiplicity					0.070
Unifocal	7	15	7	7	
Multifocal	1	6	7	0	
Microvessel invasion					0.017 *
Absent	7	12	6	2	
Present	1	9	8	5	
Portal vein invasion					0.076
Absent	8	18	10	5	
Present	0	3	4	2	
Cirrhosis					0.393
Absent	5	7	6	2	
Present	3	14	8	5	
Hepatitis virus					0.005 *
Absent	3	0	0	0	
Present	5	21	14	7	
Edmondson–Steiner grade					0.000 *
I	1	2	0	0	
II	6	6	2	1	
III	1	13	11	4	
IV	0	0	1	2	
Recurrence					0.977
Absent	3	15	8	3	
Present	5	6	6	4	
Metastasis					0.562
Absent	4	15	8	3	
Present	4	6	6	4	
5-year survival					0.005 *
Alive	5	9	5	2	
Dead	2	6	9	5	

^1^ hepatocellular carcinoma; ^2^ standard deviation; * statistically significant.

## Data Availability

Supporting data may be found in internal archives.

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
