# Peer review of "Parafibromin Is Highly Expressed in Hepatocellular Carcinoma and Its Expression Correlates with Poor Prognosis"

_jcm, 2022, doi:10.3390/jcm11071773_

Round 1

Author Response

I would like to thank you for your insightful comments and reviewing this manuscript.

Please, see the attachment!

Reviewer 2 Report

The authors present their results on the possible role of parafibromin as a predictive marker for HCC development. This study included 50 patients with HCC and analyzed the expression of parafibromin in correlation with the clinical and histological variables.

The abstract should include some data and statistical significance. 

In table 1 for some parameters, it would be better to have only the data for "Present"

I would use another word instead of "expired".

The images in Figure 2 must be verified as (a) is cut or overlapped by (b).

In the Discussions, the results should not be repeated but only discussed in correlation with other studies. It would be better to include a paragraph on the clinical significance of the data presented.

The Conclusions should include only statements based on the data presented. The other statements could be replaced in Discussions.

Author Response

I would like to thank you for your insightful comments and reviewing this manuscript.

please, see the attachment!
